# Improved calculation method for dry modal analysis of four-stage centrifugal-pump rotor system based on concentrated-mass method

**Jingkuan Li**[1]☯, **Yanxia Wei**[2]☯, **Hongbin Gao**📷[2]☯*, **Xiaomei Song**[2]☯, **Zhuofan Jia**[1]☯

**1** School of Electric Power and Architecture, Shanxi University, Taiyuan, PR China, **2** School of Automation and Software, Shanxi University, Taiyuan, PR China

☯ These authors contributed equally to this work.

* ghb9420@163.com

**Data Availability Statement:** All relevant data are within the article and its Supporting Information files.

## Abstract

To improve the accuracy of modal analysis for a four-stage centrifugal-pump rotor system with a balancing disc based on the concentrated-mass analytical method, a simplified concentrated mass mathematical model and an ANSYS simulation model are established. The results from these two models are compared to determine factors that cause significant differences in the mode shapes. Subsequently, an optimized mathematical model based on the corrected mass moment of an inertia matrix and stiffness correction coefficients is proposed, and the effectiveness of this optimized mathematical model is validated using a four-stage centrifugal pump with back blades. The results show that the natural frequencies obtained from the ANSYS simulations are consistently higher than those obtained using the analytical method. The simplification of the moment of inertia at the impeller and balancing disc contributes primarily to the calculated errors. The optimized mathematical model reduces the errors in the natural frequencies from 12.96%, 12.13%, 9.96%, 5.85%, and 8.74% to 2.45%, 1.56%, 0.65%, 5.34%, and 2.28%, respectively. The optimization of natural frequencies offers better performance at lower-order modes, whereas its effects on higher-order modes are less significant. The optimization method is applicable to centrifugal pumps with back blades and reduces the error in theoretical calculations, based on reductions in the concentrated mass from 13.11%, 12.85%, 9.91%, and 7.2% to 3.7%, 3.86%, 0.57%, and 2.87%, respectively, thus further confirming the feasibility of the optimized model design.

## 1. Introduction

### 1.1 Background

Multistage centrifugal pumps are characterized by a high head, large flow, high efficiency, and good stability, thus rendering them widely used in various fields such as wastewater treatment, sludge transport, and irrigation [1–4]. However, owing to the complex structure of multistage centrifugal pumps, stringent requirements are imposed to their design, operation, and

**Funding:** The author(s) received no specific funding for this work.

**Competing interests:** The authors have declared that no competing interests exist.

maintenance. A substantial load on the rotor system results in large vibration amplitudes [5]. Prolonged vibrations can cause fatigue damage, and resonance occurs when the vibration frequency matches the natural structural frequency, which severely affects the safety and stability of the overall system [6–9].

Modal analysis serves as the foundation for dynamic analysis by investigating the vibrational and modal characteristics of rotor systems [10–11]. It establishes a theoretical basis for reducing vibrations, avoiding resonance, ensuring safe operation, and performing dynamic analysis [12–14]. Additionally, it provides theoretical value in ensuring equipment design, predicting the operational state, and diagnosing faults in an entire system [15–17].

## 1.2 Literature review

The modal characteristics of rotor systems have been investigated extensively. Yanwei et al. [18] performed numerical simulations to investigate the effect of the curvature radii of front and rear cover plates on the natural frequencies of a high-speed centrifugal-pump impeller. They discovered that an appropriate increase in the curvature radius of the front cover plate of the impeller enhanced the modal frequencies of the centrifugal-pump impeller as well as its dynamic characteristics. Yongfei et al. [19] conducted a modal analysis on a self-balancing multistage centrifugal-pump rotor with an ultralow-specific speed using the ANSYS Workbench. They analyzed the natural frequencies of pump rotors under three conditions: a dry state without flow-field prestress, a dry state with flow-field prestress, and a wet state with flow-field prestress. The first eight mode shapes of the pump rotor were extracted and analyzed. Wenchao et al. [20] used an axial-force rotating shaft model to solve for the lateral natural frequencies using Dunkerley's method. The results were compared with finite-element method results for the first five bending vibration frequencies of the rotating shaft. The results indicated that axial pressure reduced the natural frequencies of the rotor system, whereas axial tension increased them, which significantly affected the low-order natural frequencies. Xuejun et al. [21] conducted modal simulations on a long-axis centrifugal-pump rotor and analyzed its first eight mode shapes. They concluded that the axial frequency of the pump rotor was significantly lower than the first natural frequency. The frequencies of the pump blades and guide vanes were between the second and third natural frequencies, which avoided resonance by more than 10%. Oza et al. [22] conducted a modal analysis on the radial-flow impeller of a centrifugal pump. They compared the first 20 natural frequencies extracted from an ANSYS simulation with experiment results and discovered consistency between the two sets of results, which validated the finite-element model.

Recently, scholars have performed modal analysis to detect structural damage. Chinka et al. [23] investigated a novel beam-structure damage-detection method using modal parameters. They observed that the frequency decreased when the crack depth increased at any point on the beam except for the nodes. This method accurately assessed crack location and depth. Ulriksen et al. [24] applied a previously proposed modal and wavelet analysis-based damage identification method to a wind turbine blade. Gillich et al. [25] proposed a new method based on natural-frequency changes that can detect damages in beam-like structures. They assessed the location and severity of those damages while considering the particular manner in which the natural frequencies of the weak-axis bending vibration modes changed owing to discontinuities. The application of modal fault diagnosis has broad research prospects.

Other scholars have investigated the effects of different structural materials on natural frequencies and mode shapes. For example, Pingulkar et al. [26] numerically calculated the natural frequencies and mode shapes of a cantilever centrifugal pump composed of glass-fiber- and carbon-fiber-reinforced polymer composites. They investigated the effects of changes in the

matrix material, hybridization, and stacking sequence on the natural frequencies and mode shapes. The results showed that the hybridization and orientation of the outermost layer significantly affected the natural frequencies of the composite laminates. Kumar et al. [27] compared the natural frequencies of a steel-based centrifugal pump with those of a composite raft foundation centrifugal pump using the ANSYS software. These results indicated that composite materials enhanced the natural frequencies of the system.

The concentrated-mass method used in this study has been similarly used in other studies. Licai [28] used a mining shearer gearbox test bench as the research object, utilized the equivalent-mass method to consider the gravity of the gearbox as a prestress, and conducted a prestressed modal analysis on the test bench. Torabi et al. [29] used the VIM and considered both shear deformation and rotatory inertia in investigating the transverse vibration of a nonlinear Timoshenko beam supporting a concentrated mass oscillating with a large amplitude. They presented a suitable trend for determining the Lagrange multiplier by considering the effects of concentrated mass on natural frequencies in linear and nonlinear states. Xuan et al. [30] investigated the strong nonlinear vibration of a cantilever beam system with a concentrated mass at an intermediate position controlled by displacement and velocity time delay.

Although modal analysis has been investigated extensively, including analyses of the modal characteristics of the aforementioned centrifugal-pump rotor systems, researchers have focused primarily on mesh partitioning, material types, and external factors. Studies pertaining to the optimization of mathematical models for centrifugal-pump rotor systems are limited.

Hongbin et al. [31, 32] performed investigations on both double-support and cantilever four-stage centrifugal-pump rotor systems. They employed different simplified mathematical models and ANSYS models to solve for the natural frequencies and compared the results of the two calculation methods. An optimized mathematical model was proposed. Additionally, the effect of impeller-mass eccentricity on the modes was analyzed. This study differed from those mentioned earlier in terms of the research objects and optimization models.

### 1.3 Contribution of current study

Modal analysis is a crucial basis for analyzing vibration issues in centrifugal pumps and is an essential factor in the design of dynamic devices [33]. This study focuses on the rotor system of a four-stage centrifugal pump with a balancing disc fixed using a simple beam support. In this study, the modal characteristics of the rotor system are analyzed using the concentrated-mass method. Subsequently, the results are compared with those obtained via ANSYS simulations, where variations in the natural frequencies and mode shapes are examined to determine the potential reasons contributing to the discrepancies observed as well as to identify the influencing factors. Additionally, an optimized mathematical model is proposed to reduce the errors in the modal results obtained via analytical calculations for a four-stage centrifugal-pump rotor system. Furthermore, the optimization method is applied to a four-stage centrifugal pump with blades for axial-force balance, which yielded favorable modal-analysis results. This optimization method not only provides essential insights for centrifugal-pump design but also offers valuable reference for fault diagnosis.

## 2. Mathematical model

When conducting a dynamic analysis using the finite-element method, the motion equations for a multi-degree-of-freedom linear structural system can be expressed as follows [34]:

$$M\{\ddot{x}(t)\} + C\{\dot{x}(t)\} + K\{x(t)\} = \{F(t)\}, \tag{1}$$

where $M$, $C$, and $K$ denote the mass, damping, and stiffness matrices of the system,

respectively. They are symmetric matrices of order $n$, which is the number of degrees-of-freedom of the system. Meanwhile, $\{x(t)\}$ is the displacement vector of the system, and its first and second derivatives are the velocity and acceleration vectors, respectively; and $\{F(t)\}$ denotes the external load vector. The modal characteristics represent the intrinsic properties of the system and can be regarded as undamped free vibrations, i.e., $C = 0$ and $\{F(t)\} = 0$. Therefore, in the absence of damping, the equation of motion for the free vibration of the system can be expressed as follows [35]:

$$M\{\ddot{x}(t)\} + K\{x(t)\} = 0. \tag{2}$$

To solve this equation, the form of the solution is assumed as follows [35]:

$$x(t) = \varphi\sin(\omega t + \theta) \tag{3}$$

By substituting Eq (3) into Eq (2), the following characteristic equation is obtained:

$$(K - \omega^2 M)\varphi = 0, \tag{4}$$

where $(K-\omega^2 M)$ is the characteristic matrix of the equation and $\varphi$ is the mode shape vector of the system. The modal analysis involves solving for the natural frequency $\omega_i$ and vibration vector $\varphi_i$. Because the mode shapes at various points in the free vibration are not zero, they satisfy Eq (5).

$$|K - \omega^2 M| = 0 \tag{5}$$

Rearranging Eq (5) yields

$$\left|\frac{1}{\omega^2}I - K^{-1}M\right| = 0 \tag{6}$$

In the equation above, $K^{-1}$ is referred to as the flexibility matrix of the system. Using Eq (6), a series of $1/\omega_i^2$ and $\varphi_i$ values can be solved. Each $1/\omega_i^2$ corresponds to a set of characteristic vectors $\varphi_i$, and each eigenvalue and its corresponding eigenvector determine one mode of free vibration of the structure. The relationship between the $i$-th eigenvalue and $i$-th natural frequency is expressed as follows [35]:

$$f_i = \frac{\omega_i}{2\pi} \tag{7}$$

The model of a four-stage centrifugal-pump rotor system that uses a balancing disc comprises four impellers and one balancing disc, as shown in Fig 1. The dimensions and structure of the pump are as follows: the total length of the pump shaft is $l$, the distance between impeller 1 and the left shaft end is $a$, the distance between impeller 4 and the balancing disc is $b$, the length of the balancing disc to the right shaft end is $c$, and the axial distances between the impellers are equal. Therefore, the axial distance between the two impellers is expressed as

$$s = \frac{l - a - b - c}{3} \tag{8}$$

The pump shaft is supported at both ends of the casing. This support structure is characteristic of a simply supported beam structure. Therefore, the radial displacements of the impeller and balancing disc can be calculated using the beam deformation. This allows one to determine the relative radial displacement at each impeller arising from the radial force applied to the shaft by each impeller, i.e., the flexibility influence coefficient. The flexibility influence

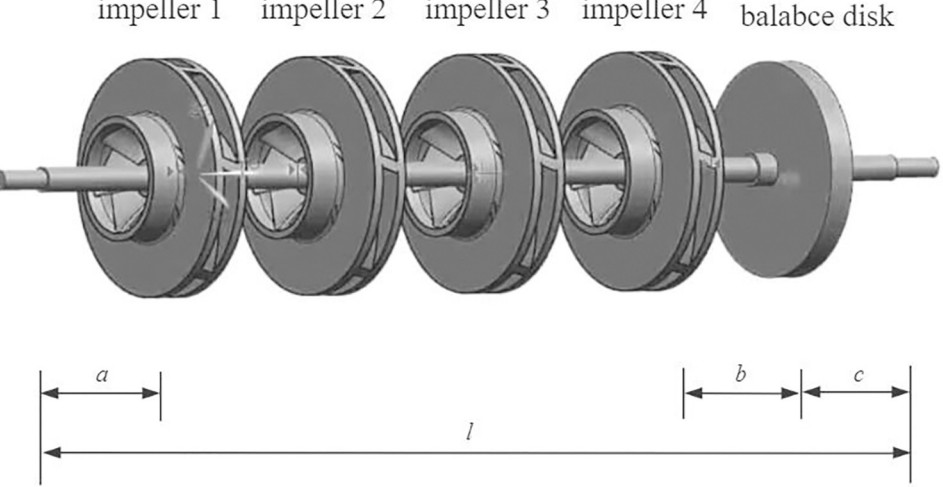

**Fig 1. Schematic diagram of four-stage centrifugal-pump rotor system with balance disc.**

coefficient represents the displacement generated under the action of a unit force and is denoted by $\delta_{ij}$, where $j$ and $i$ indicate the positions where force is applied on the coordinate direction of the $j$-th and $i$-th mass blocks, respectively. The matrix comprising the flexibility influence coefficients is the inverse of the stiffness matrix. The deformation level of the beam can be assessed by calculating and analyzing the flexibility influence coefficients of the beam, thus allowing one to determine the stability and safety of the structure. Based on the principles of material mechanics and the bending equation of the beam, the flexibility influence coefficients of impeller 1 at the positions of impellers 1, 2, 3, and 4 and the balancing disc are expressed as follows:

$$\delta_{11} = \frac{a^2(l-a)^2}{3EIl} \tag{9A}$$

$$\delta_{12} = \frac{(l-a)}{6EIl}\left\{\frac{l}{l-a}(x-a)^3 + \left[l^2-(l-a)^2\right]x - x^3\right\}\Big|_{x=\frac{l+2a-b-c}{3}} \tag{9B}$$

$$\delta_{13} = \frac{(l-a)}{6EIl}\left\{\frac{l}{l-a}(x-a)^3 + \left[l^2-(l-a)^2\right]x - x^3\right\}\Big|_{x=\frac{2l+a-2b-2c}{3}} \tag{9C}$$

$$\delta_{14} = \frac{(l-a)}{6EIl}\left\{\frac{l}{l-a}(x-a)^3 + \left[l^2-(l-a)^2\right]x - x^3\right\}\Big|_{x=l-b-c} \tag{9D}$$

$$\delta_{15} = \frac{(l-a)}{6EIl}\left\{\frac{l}{l-a}(x-a)^3 + \left[l^2-(l-a)^2\right]x - x^3\right\}\Big|_{x=l-c} \tag{9E}$$

The flexibility influence coefficients for impeller 2 at impellers 1, 2, 3, and 4 and the balancing disc are expressed as follows:

$$\delta_{21} = \frac{a(2l-2a+b+c)}{162EIl}\left[9l^2 - 9a^2 - (2l-2a+b+c)^2\right] \tag{10A}$$

$$\delta_{22} = \frac{(l+2a-b-c)(2l-2a+b+c)}{486EIl} \left[9l^2 - (l+2a-b-c)^2 - (2l-2a+b+c)^2\right] \quad (10B)$$

$$\delta_{23} = \frac{(2l-2a+b+c)}{18EIl} \left\{ \frac{3l}{2l-2a+b+c} \left[x - \frac{(l+2a-b-c)}{3}\right]^3 + \left[l^2 - \frac{(2l-2a+b+c)^2}{9}\right]x - x^3 \right\} \Big|_{x=\frac{2l+a-2b-2c}{3}} \quad (10C)$$

$$\delta_{24} = \frac{(2l-2a+b+c)}{18EIl} \left\{ \frac{3l}{2l-2a+b+c} \left[x - \frac{(l+2a-b-c)}{3}\right]^3 + \left[l^2 - \frac{(2l-2a+b+c)^2}{9}\right]x - x^3 \right\} \Big|_{x=l-b-c} \quad (10D)$$

$$\delta_{25} = \frac{(2l-2a+b+c)}{18EIl} \left\{ \frac{3l}{2l-2a+b+c} \left[x - \frac{(l+2a-b-c)}{3}\right]^3 + \left[l^2 - \frac{(2l-2a+b+c)^2}{9}\right]x - x^3 \right\} \Big|_{x=l-c} \quad (10E)$$

Similarly, the flexibility influence coefficients for impeller 3 at impellers 1, 2, 3, and 4 and the balancing disc are expressed as follows:

$$\delta_{31} = \frac{a(l-2a+2b+2c)}{162EIl} \left[9l^2 - 9a^2 - (l-a+2b+2c)^2\right] \quad (11A)$$

$$\delta_{32} = \frac{(l+2a-b-c)(l-a+2b+2c)}{486EIl} \left[9l^2 - (l+2a-b-c)^2 - (l-a+2b+2c)^2\right] \quad (11B)$$

$$\delta_{33} = \frac{(2l+a-2b-2c)(l-a+2b+2c)}{486EIl} \left[9l^2 - (2l+a-2b-2c)^2 - (l-a+2b+2c)^2\right] \quad (11C)$$

$$\delta_{34} = \frac{(l-a+2b+2c)}{18EIl} \left\{ \frac{3l}{l-a+2b+2c} \left[x - \frac{(2l+a-2b-2c)^3}{3}\right]^3 + \left[l^2 - \frac{(l-a+2b+2c)^2}{9}\right]x - x^3 \right\} \Big|_{x=l-b-c} \quad (11D)$$

$$\delta_{35} = \frac{(l-a+2b+2c)}{18EIl} \left\{ \frac{3l}{l-a+2b+2c} \left[x - \frac{(2l+a-2b-2c)^3}{3}\right]^3 + \left[l^2 - \frac{(l-a+2b+2c)^2}{9}\right]x - x^3 \right\} \Big|_{x=l-c} \quad (11E)$$

The flexibility influence coefficients for impeller 4 at impellers 1, 2, 3, and 4 and the balancing disc are expressed as follows:

$$\delta_{41} = \frac{a(b+c)}{6EIl} \left[l^2 - a^2 - (b+c)^2\right] \quad (12A)$$

$$\delta_{42} = \frac{(l+2a-b-c)(b+c)}{18EIl} \left[l^2 - \frac{(l+2a-b-c)^2}{9} - (b+c)^2\right] \quad (12B)$$

$$\delta_{43} = \frac{(2l+a-2b-2c)(b+c)}{18EIl} \left[l^2 - \frac{(2l+a-2b-2c)^2}{9} - (b+c)^2\right] \quad (12C)$$

$$\delta_{44} = \frac{(l-b-c)(b+c)}{6EIl} \left[l^2 - (l-b-c)^2 - (b+c)^2\right] \quad (12D)$$

$$\delta_{45} = \frac{(b+c)}{6EIl} \left\{ \frac{l}{b+c}[x-(l-b-c)]^3 + [l^2-(b+c)^2]x - x^3 \right\}\Big|_{x=l-c} \tag{12E}$$

Finally, the flexibility influence coefficients for the balancing disc at impellers 1, 2, 3, and 4 and the balancing disc itself are expressed as follows:

$$\delta_{51} = \frac{ac}{6EIl}[l^2 - a^2 - c^2] \tag{13A}$$

$$\delta_{52} = \frac{(l+2a-b-c)c}{18EIl} \left[ l^2 - \frac{(l+2a-b-c)^2}{9} - c^2 \right] \tag{13B}$$

$$\delta_{53} = \frac{(2l+a-2b-2c)c}{18EIl} \left[ l^2 - \frac{(2l+a-2b-2c)^2}{9} - c^2 \right] \tag{13C}$$

$$\delta_{54} = \frac{(l-b-c)c}{6EIl} \left[ l^2 - (l-b-c)^2 - c^2 \right] \tag{13D}$$

$$\delta_{55} = \frac{(l-c)c}{6EIl} \left[ l^2 - (l-c)^2 - c^2 \right], \tag{13E}$$

where $EI$ denotes the sectional stiffness coefficient of the beam. Therefore, based on Eqs (9)–(13), the flexibility matrix of the system can be obtained as follows:

$$K^{-1} = \begin{bmatrix} \delta_{11} & \delta_{12} & \delta_{13} & \delta_{14} & \delta_{15} \\ \delta_{21} & \delta_{22} & \delta_{23} & \delta_{24} & \delta_{25} \\ \delta_{31} & \delta_{32} & \delta_{33} & \delta_{34} & \delta_{35} \\ \delta_{41} & \delta_{42} & \delta_{43} & \delta_{44} & \delta_{45} \\ \delta_{51} & \delta_{52} & \delta_{53} & \delta_{54} & \delta_{55} \end{bmatrix} \tag{14}$$

By assuming that the pump shaft, impellers, and balancing disc constitute a series of discretely distributed concentrated mass points, their vibration modes are mutually independent and no coupling effect exists. Therefore, the concentrated-mass method can be effectively applied to the modal analysis of this rotor system.

By partitioning the entire pump shaft into five concentrated mass segments, the mass at each concentrated mass point can be expressed as follows:

$$m_1 = m_i + \pi r^2 (a+s)\rho \tag{15A}$$

$$m_2 = m_i + \pi r^2 s\rho \tag{15B}$$

$$m_3 = m_i + \pi r^2 s\rho \tag{15C}$$

$$m_4 = m_i + \pi r^2 \frac{s+b}{2}\rho \tag{15D}$$

$$m_5 = m_i + \pi r^2 (b+c)\rho, \tag{15E}$$

**Table 1. Natural frequencies of each order obtained from analytical calculations.**

| Order | First | Second | Third | Fourth | Fifth |
|---|---|---|---|---|---|
| Natural frequency (Hz) | 21.74 | 83.94 | 181.61 | 304.2 | 406.66 |

where $m_i$ and $m_j$ represent the masses of the impeller (kg) and balancing disc (kg), respectively; $m_1$ includes the mass of the shaft segment from the left bearing end to impeller 2 (kg); $m_2$ is the mass of the shaft segment between impellers 1 and 2 and between impellers 2 and 3 (kg); $m_3$ is the mass of the shaft segment between impellers 2 and 3 and between impellers 3 and 4 (kg); $m_4$ is the mass of the shaft segment between impellers 3 and 4 and between impeller 4 and the balancing disc, i.e., $m_2 = m_3$, (kg); and $m_5$ is the mass of the shaft segment between impeller 4 and the right-bearing end (kg).

Therefore, the mass matrix $[M]$ of the dynamic equation of the four-stage pump rotor system is as follows:

$$[M] = \begin{bmatrix} m_i + \pi r^2(a+s)\rho & 0 & 0 & 0 & 0 \\ 0 & m_i + \pi r^2 s\rho & 0 & 0 & 0 \\ 0 & 0 & m_i + \pi r^2 s\rho & 0 & 0 \\ 0 & 0 & 0 & m_i + \pi r^2 \dfrac{s+b}{2}\rho & 0 \\ 0 & 0 & 0 & 0 & m_i + \pi r^2(b+c)\rho \end{bmatrix} \quad (16)$$

The natural frequencies of the four-stage centrifugal-pump rotor system can be obtained by substituting Eqs (14) and (16) into Eq (6). In this pump, $l = 0.8\ m$, $a = 0.106\ m$, $s = 0.15\ m$, $b = 0.142\ m$, $c = 0.102\ m$, and $\rho = 7850\ kg/m^3$; additionally, the shaft section diameter is 0.023 $m$, the elastic modulus $E$ is $2.1{\times}10^{11}\ Pa$, and the section moment of inertia is calculated as $I = \pi r^4/4$, which yields $I = 1.350 \times 10^{-8}\ m^4$. The masses of the impeller, balancing disc, and pump shaft in the system are 4.776, 5.8, and 2.738 $kg$, respectively. Using these parameters, the natural frequencies of each order of the rotor system can be calculated, as listed in Table 1.

By substituting the flexibility matrix, mass matrix, and the values of the natural frequencies for each order into the characteristic equation presented in Eq (4), the normalized mode shape matrix $\varphi$ for the multistage centrifugal-pump rotor system can be obtained. The results are shown in Eq (17).

$$\varphi = \begin{bmatrix} 1 & 1 & 1 & 1 & 1 \\ 2.0879 & 1.2390 & 0.1419 & -0.8734 & -1.4907 \\ 2.4758 & 0.0016 & -1.0814 & 0.0320 & 1.6733 \\ 2.0322 & -1.2178 & 0.0976 & 0.8533 & -1.6166 \\ 0.9723 & -0.9753 & 0.9172 & -0.8612 & 0.9684 \end{bmatrix} \quad (17)$$

Subsequently, the mode shapes for each order of the four-stage centrifugal-pump rotor system can be obtained (Fig 2). Fig 2 shows that under the first three natural frequencies, the deformation directions of impellers 1 and 2 and the deformation directions of impeller 4 and the balancing disc were consistent. At the last two natural frequencies, the deformation directions of impellers 1 and 2 and those of impeller 4 and the balancing disc were opposite. Among the five natural frequencies, the radial deformation of impeller 3 was the most severe. At the fifth natural frequency, the entire system deformed the most frequently, and the deformation directions between adjacent mass blocks were opposite.

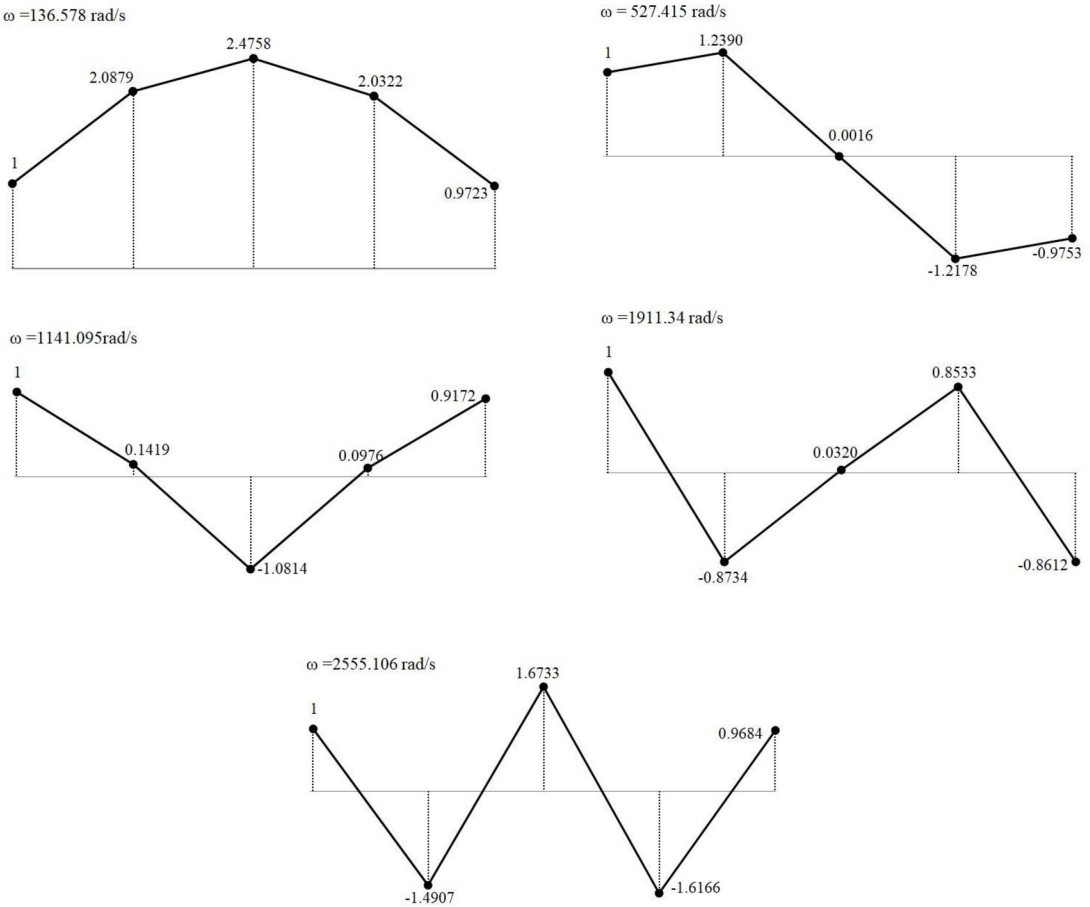

**Fig 2. Vibration mode diagram of each order calculated using analytical method.** (a) First-order modal shape. (b) Second-order modal shape. (c) Third-order modal shape. (d) Fourth-order modal shape. (e) Fifth-order modal shape.

## 3. ANSYS simulation and analysis

In this study, the ANSYS Workbench collaborative simulation platform was used to simulate the modal behavior of a multistage centrifugal pump in the free state [36]. During the simulation, the rotor system must be meshed. Therefore, this analysis can be regarded as a microscopic, computer-based analytical solution using the concentrated-mass method. Additionally, based on numerous research findings, the numerically obtained modal behavior in beam-like structures typically coincides with experimental results [37–38]. Therefore, the simulation results obtained from ANSYS Workbench were utilized as reference values for the actual results in this study.

A four-stage centrifugal-pump rotor system with a balancing disc, shown in Fig 1, was selected as the research subject in this study. The core components for system modeling were an impeller, a pump shaft, and a balancing disc. Three-dimensional (3D) modeling was performed using the UG software and PCAD pump hydraulic design software. Modal analysis of the multistage centrifugal-pump system was performed using the Modal module in the ANSYS Workbench software to determine the natural frequencies and mode shapes [39–41]. In this study, the entire model was meshed with a mesh size of 7 *mm* [42]. The entire model was segmented into 183,431 nodes and 102,154 elements, and the simulation results are shown in Fig 3.

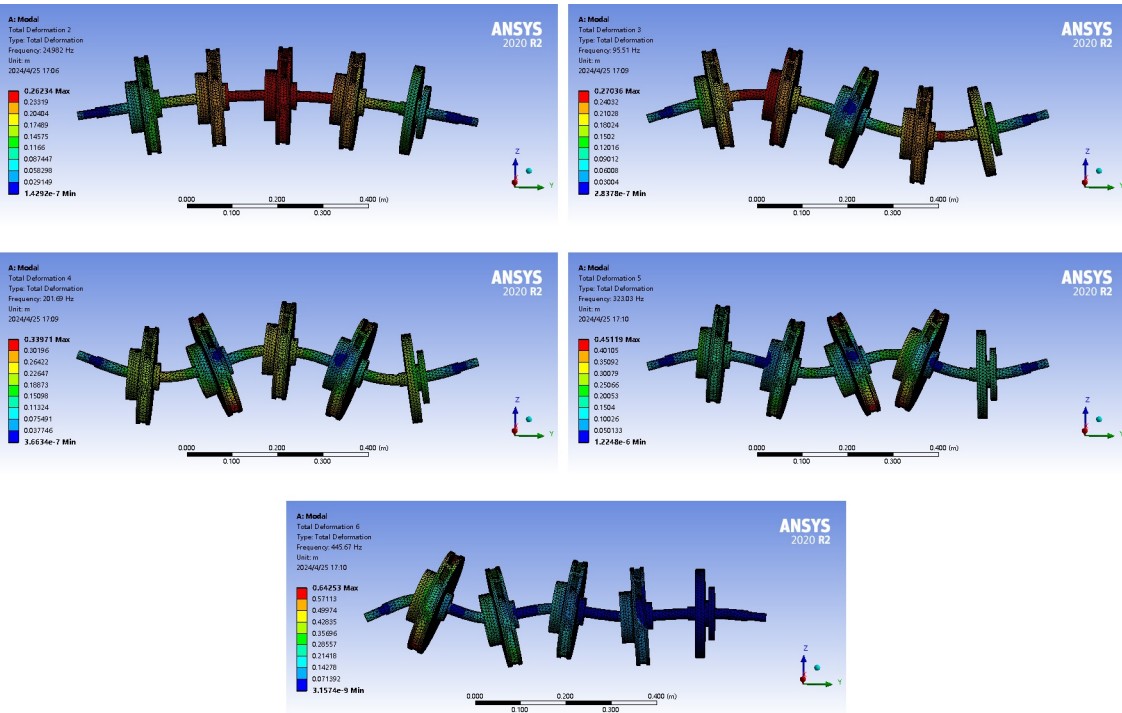

**Fig 3. ANSYS simulation modal-shape diagram of centrifugal pump with balance plate.** (a) First-order mode. (b) Second-order mode. (c) Third-order mode. (d) Fourth-order mode. (e) Fifth-order mode.

Based on the animation demonstration in the Workbench modal analysis, the mode shapes for the fifth order exhibited bending deformation in the Z-direction. A comparison of the natural frequencies obtained from the analytical calculations of the mathematical model and those obtained from the ANSYS simulation is presented in Table 2.

As shown in Table 2, the natural frequencies obtained using the concentrated-mass method were consistently lower than those obtained from the ANSYS simulation, with a certain degree of error (maximum error of 12.96%).

## 4. Optimized mathematical model

In a centrifugal-pump rotor system, the inherent frequency and mode shapes are related to the structural characteristics and material properties, such as the elastic modulus and density of the material, and the mass and stiffness of the system. The inherent frequencies of the rotor system differed significantly from the results of 3D simulations, which is attributable to the following reasons: (1) repeated consideration of the axial spacing mass between impellers 1 and 2 as well as between impeller 4 and the balance disc; and (2) non-consideration of the effect of

**Table 2. Comparison between analytical calculation and ANSYS simulation results.**

| Modal Order | Calculated value (Hz) | Simulated value (Hz) | Percent difference (%) |
|---|---|---|---|
| First Order | 21.74 | 24.98 | 12.96 |
| Second Order | 83.74 | 95.51 | 12.13 |
| Third Order | 181.61 | 201.69 | 9.96 |
| Fourth Order | 304.2 | 323.03 | 5.85 |
| Fifth Order | 406.66 | 445.67 | 8.74 |

inertia moments at the impeller and balance-disc locations when calculating the cross-sectional moment of inertia of the system. Based on this analysis, efforts were expended to reduce the error between the inherent frequencies obtained from the analytical calculations and the results of 3D simulations. Subsequently, an optimized analytical method for mathematical modeling is proposed.

Considering that the segments supported by the left and right bearings will not deform, one must reduce the mass of the segments between the left bearing and impeller 1 as well as that between the balancing disc and right bearing, both of which contribute to the calculation of the dynamic characteristics of the system. The corrected values at the concentrated mass points are obtained as follows:

$$
\begin{aligned}
m_1' &= m_i + \pi r^2 \frac{a+s}{2}\rho \\
m_2' &= m_3' = m_i + \pi r^2 s\rho \\
m_4' &= m_i + \pi r^2 \frac{b+s}{2}\rho \\
m_5' &= m_i + \pi r^2 \frac{b+c}{2}\rho
\end{aligned}
\tag{18}
$$

$m_1'$ includes the mass of the shaft segment from the left bearing end to that between impellers 2 (kg); $m_2'$ is the mass of the shaft segment between impellers 1 and 2 and between impellers 2 and 3 (kg); $m_3'$ is the mass of the shaft segment between impellers 2 and 3 and between impellers 3 and 4 (kg); $m_4'$ is the mass of the shaft segment between impellers 3 and 4 and between impeller 4 and the balancing disc, i.e., $m_3 = m_4$ (kg); and $m_5'$ is the mass of the shaft segment between impeller 4 and the right-bearing end (kg).

The stiffness $EI$ of the shaft section is composed of the bending stiffnesses of the pump shaft and impeller.

$$
EI = (EI)_a + (EI)_s,
\tag{19}
$$

where $(EI)_a$ and $(EI)_s$ represent the bending stiffness of the pump shaft and impeller, respectively. Because of the much smaller cross-sectional area of the impeller blades compared with those of the front and rear cover plates, only the bending stiffness of the front and rear cover plates is considered at the impeller. Subsequently, an additional bending stiffness correction factor $\alpha$ is introduced, as shown in Eq (20).

$$
\alpha = \frac{1}{\frac{T-t_1}{T} + \frac{t_2+t_3}{T}\left(\frac{d}{D}\right)^4}
\tag{20}
$$

This coefficient represents the ratio of the corrected and uncorrected stiffness values. Here, $T$ is the effective support length of the rotor system. For the impeller, $t_1$ is the thickness of the impeller hub; $t_2$ and $t_3$ are the thicknesses of the front and rear cover plates of the impeller, respectively; $d$ is the diameter of the pump shaft; and $D$ is the hub diameter. For the balance disc, $t_1$ is the thickness of the balance-disc hub; and $t_2$ and $t_3$ are the thicknesses of the left and right plates, respectively. Based on the pump dimensions, the values at the five concentrated-mass points were 1.136, 1.256, 1.256, 1.265, and 1.187. Based on the stiffness correction factor, one can infer that the larger the value of $T$, the smaller is the correction factor $\alpha$. Impeller 4 exerted the most significant effect on the stiffness of the rotor system, whereas impeller 1 exerted the least effect. The effective length selected for the concentrated-mass segment directly affected the magnitude of $\alpha$.

The modified-mass and flexibility matrices yielded the dynamic matrix $K^{-1}M$ of the optimized mathematical model as follows:

$$K^{-1}M = \begin{bmatrix} \dfrac{m'_1\delta_{11}}{\alpha_1} & \dfrac{m'_2\delta_{12}}{\alpha_1} & \dfrac{m'_3\delta_{13}}{\alpha_1} & \dfrac{m'_4\delta_{14}}{\alpha_1} & \dfrac{m'_5\delta_{15}}{\alpha_1} \\[2mm] \dfrac{m'_1\delta_{11}}{\alpha_2} & \dfrac{m'_2\delta_{12}}{\alpha_2} & \dfrac{m'_3\delta_{13}}{\alpha_2} & \dfrac{m'_4\delta_{14}}{\alpha_2} & \dfrac{m'_5\delta_{15}}{\alpha_2} \\[2mm] \dfrac{m'_1\delta_{21}}{\alpha_3} & \dfrac{m'_2\delta_{22}}{\alpha_3} & \dfrac{m'_3\delta_{23}}{\alpha_3} & \dfrac{m'_4\delta_{24}}{\alpha_3} & \dfrac{m'_5\delta_{25}}{\alpha_3} \\[2mm] \dfrac{m'_1\delta_{31}}{\alpha_4} & \dfrac{m'_2\delta_{32}}{\alpha_4} & \dfrac{m'_3\delta_{33}}{\alpha_4} & \dfrac{m'_4\delta_{34}}{\alpha_4} & \dfrac{m'_5\delta_{35}}{\alpha_4} \\[2mm] \dfrac{m'_1\delta_{41}}{\alpha_5} & \dfrac{m'_2\delta_{42}}{\alpha_5} & \dfrac{m'_3\delta_{43}}{\alpha_5} & \dfrac{m'_4\delta_{44}}{\alpha_5} & \dfrac{m'_5\delta_{45}}{\alpha_5} \end{bmatrix} \qquad (21)$$

Substituting Eq (21) into Eq (6) and combining the parameters of the previous system yielded the natural frequencies for the optimized mathematical model, i.e., 24.37, 94.02, 203.00, 340.29, and 455.82 Hz.

The natural frequencies obtained from the pre-optimized analytical calculations were compared with the results obtained from the post-optimized mathematical model by adopting the ANSYS simulation results in Table 2 as the reference benchmark, as shown in Fig 4. After optimization, the errors in the natural frequencies for each order, the system mass, and the inertia decreased. The acceleration in the response spe ed of the system increased the frequency, which yielded results that were more similar to the simulation results. Therefore, one can conclude that the optimized design method mentioned above is feasible and can, to a certain extent, reduce the natural frequencies calculated using the mathematical model. Substituting

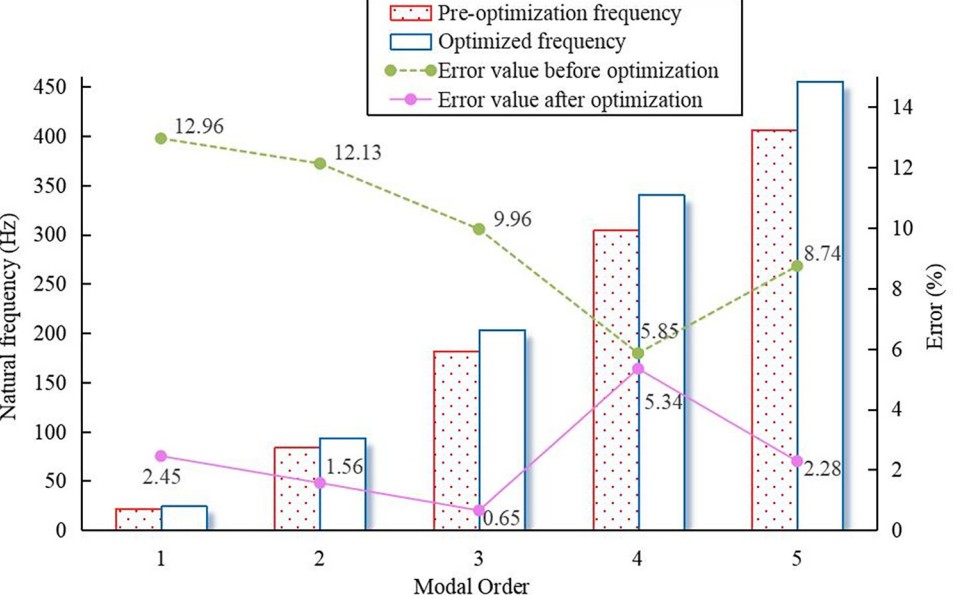

**Fig 4. Comparison between analytical calculations and optimization results.**

xxx into Eq (4) yields the normalized mode shape vectors after optimization, as follows:

$$\varphi' = \begin{bmatrix} 1 & 1 & 1 & 1 & 1 \\ 1.8873 & 1.1141 & 0.1138 & -0.8187 & -1.3897 \\ 2.2366 & -0.0081 & -0.9830 & 0.0485 & 1.5739 \\ 1.8224 & -1.0985 & 0.0979 & 0.7726 & -1.5314 \\ 0.9292 & -0.9341 & 0.8840 & -0.8404 & 0.9811 \end{bmatrix} \qquad (22)$$

The optimized mode shapes for each order are shown in Fig 5.

A comparison between Figs 2 and 5 shows that the deformation directions of each mode shape were generally consistent before and after optimization. At the first-order natural frequency, the deformation of impeller 3 was the greatest. A symmetric deformation distribution occurred at the first-, third-, and fifth-order natural frequencies. At the fifth-order natural frequency, the deformation directions between the impellers were opposite, and the deformation was maximal, thus indicating a relatively greater effect on the axial system. After optimization, the mode-shape variations of impellers 1 and 2 were more similar to the actual variations, thus indicating a favorable optimization outcome.

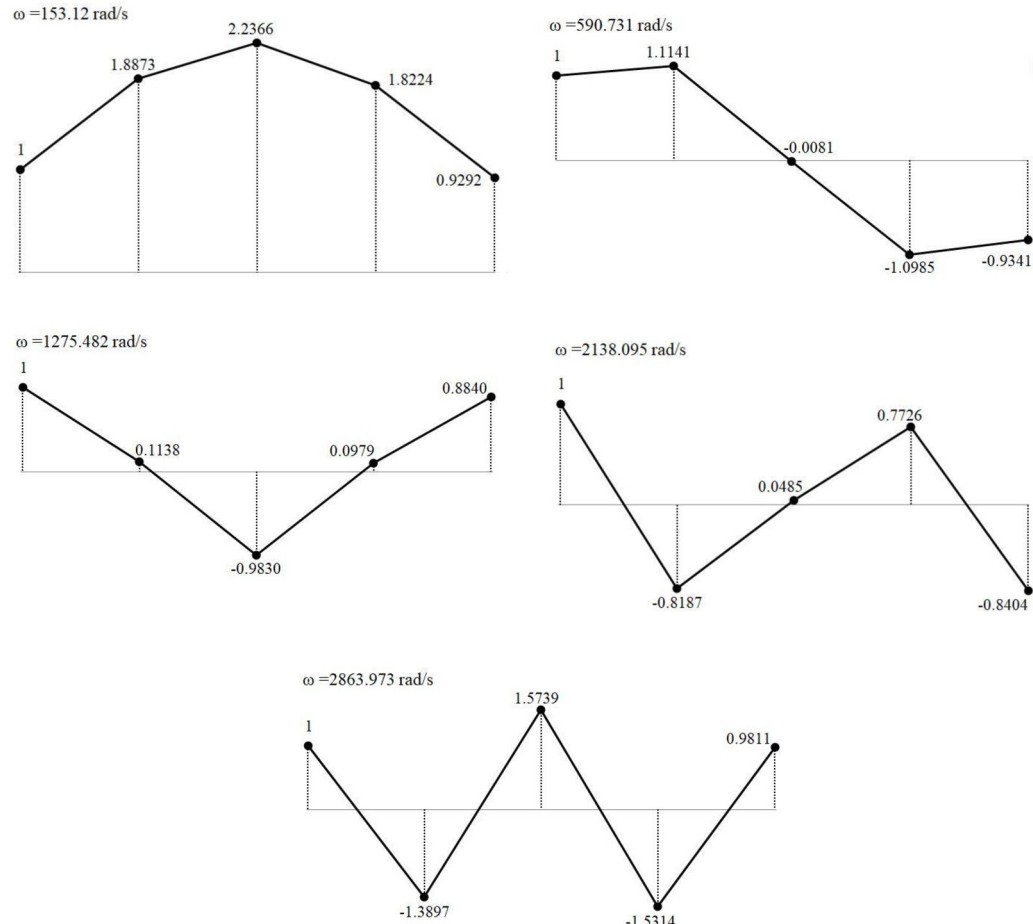

**Fig 5. Optimized mode shapes for each order.** (a) First-order modal shape. (b) Second-order modal shape. (c) Third-order modal shape. (d) Fourth-order modal shape. (e) Fifth-order modal shape.

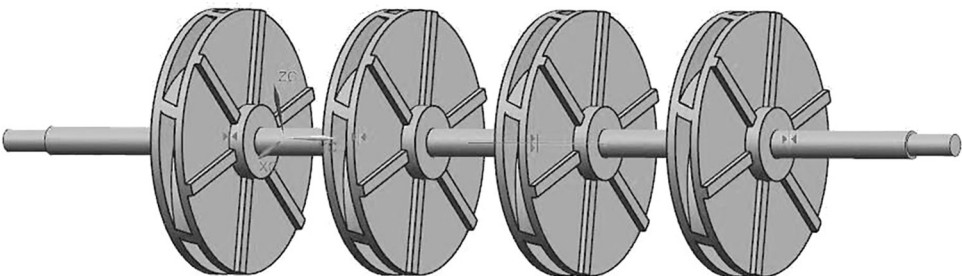

**Fig 6. Four-stage centrifugal pump with back-bladed impellers.**

## 5. Application of optimization method to back-bladed impeller of four-stage centrifugal pump

The axial force during the operation of a centrifugal pump is caused by an uneven pressure distribution on both sides of the impeller due to fluid action. Balancing axial forces is crucial in centrifugal-pump design, particularly for multistage centrifugal pumps, such as four-stage centrifugal pumps. Back blades are widely employed in multistage centrifugal pumps that do not use a balance disc as an axial-force balancing device [43]. Therefore, in this study, the improved mathematical model was applied to a four-stage centrifugal pump using back blades for axial-force balancing. The balance disc in the four-stage centrifugal pump shown in Fig 1 was removed, and the pump was modified to include impellers with back blades attached to the outer side of the back-cover plate, as illustrated in Fig 6.

The Modal module in the ANSYS Workbench software was used to perform modal analysis on the four-stage centrifugal pump with back blades to determine its natural frequencies and mode shapes. During 3D modeling, the four impellers were distributed uniformly along the pump shaft. Similar to the modal simulation analysis described in the previous section, the same meshing approach and constraint conditions were employed in the ANSYS software. The natural frequencies obtained for each order were 25.368, 98.905, 212, and 345.9 Hz. The modal shapes for each order are shown in Fig 7.

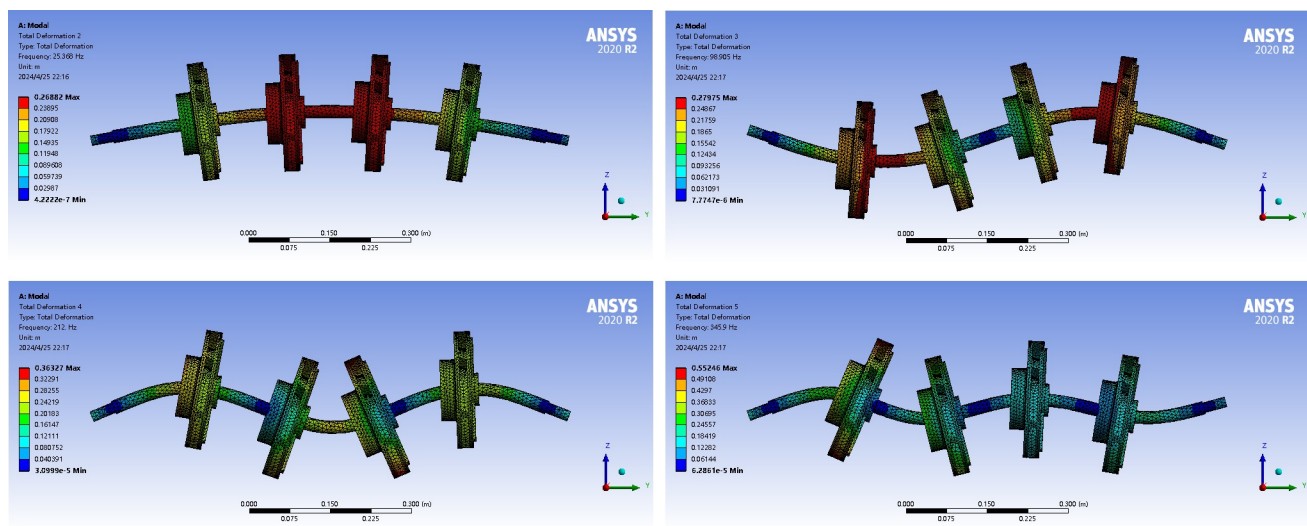

**Fig 7. Modal vibration mode diagram based on ANSYS simulation.** (a) First-order mode. (b) Second-order mode. (c) Third-order mode. (d) Fourth-order mode.

Simultaneously, the natural frequencies of the centrifugal pump were calculated using the lumped-mass method, and the mass matrix is represented as follows:

$$
M = \begin{bmatrix}
m_i + \dfrac{\pi d^2 l \rho}{10} & 0 & 0 & 0 \\[2mm]
0 & m_i + \dfrac{\pi d^2 l \rho}{20} & 0 & 0 \\[2mm]
0 & 0 & m_i + \dfrac{\pi d^2 l \rho}{20} & 0 \\[2mm]
0 & 0 & 0 & m_i + \dfrac{\pi d^2 l \rho}{10}
\end{bmatrix}
\tag{23}
$$

The stiffness matrix is expressed as

$$
K^{-1} = \frac{l^3}{3750EI}\begin{bmatrix}
32 & 45 & 40 & 23 \\
45 & 72 & 68 & 40 \\
40 & 68 & 72 & 45 \\
23 & 40 & 45 & 32
\end{bmatrix}
\tag{24}
$$

The impeller mass $m_i$ was 4.962 kg and the pump-shaft mass was 2.738 kg. By substituting the expressions above into Eq (6), the natural frequencies obtained via mass-centralized analytical calculation were 22.04, 86.20, 190.98, and 320.92 Hz. Subsequently, the improved optimization method proposed herein was employed to optimize the analytical calculation values.

$$
M' = \begin{bmatrix}
m_i + \dfrac{\pi d^2 l \rho}{20} & 0 & 0 & 0 \\[2mm]
0 & m_i + \dfrac{\pi d^2 l \rho}{20} & 0 & 0 \\[2mm]
0 & 0 & m_i + \dfrac{\pi d^2 l \rho}{20} & 0 \\[2mm]
0 & 0 & 0 & m_i + \dfrac{\pi d^2 l \rho}{20}
\end{bmatrix}
\tag{25}
$$

By substituting the values of 0.32, 0.16, 0.16, and 0.32 into Eq (20), stiffness correction factors of 1.106, 1.236, 1.236, and 1.106 were obtained, respectively. Subsequently, by substituting the corrected mass matrix and stiffness correction factors into Eq (21), optimized natural frequencies of 24.43, 95.091, 210.782, and 355.843 Hz were obtained, respectively.

Based on a comparison between the analytical calculation and optimization results shown in Fig 8, the errors in the analytical calculation values decreased from 13.11%, 12.85%, 9.91%, and 7.2% to 3.7%, 3.86%, 0.57%, and 2.87%, respectively, thus indicating the effectiveness of the optimized mathematical model.

Similarly, the vibration mode of the four-stage centrifugal pump with back blades changed in the Z-direction. The deformation directions of the first and fourth orders were consistent with those of the multistage centrifugal pump with a balancing disc, whereas those of the second and third orders were opposite. The optimization method for a multistage centrifugal pump with a balancing disc is applicable to a four-stage centrifugal pump with back blades. Compared with the multistage centrifugal pump using the balance disc as the axial-force-balance device, the four-stage centrifugal pump using the back blade as the axial-force-balance method indicated a significantly reduced overall error by adopting the improved mathematical model, as well as a favorable optimization effect.

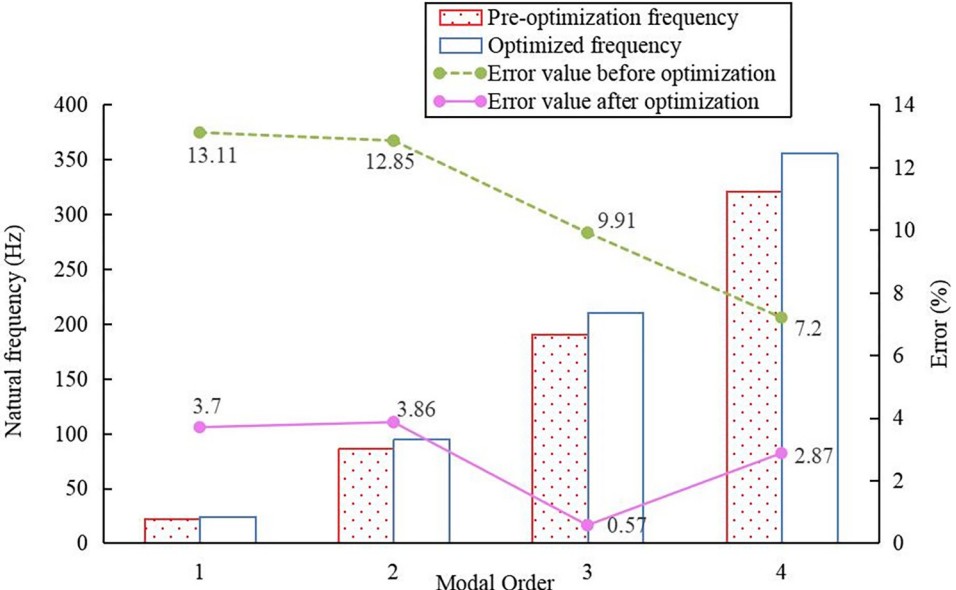

**Fig 8. Comparison between analytical calculation and optimization results of back-vane centrifugal pump.**

## 6. Conclusion

In this study, a four-stage centrifugal-pump rotor system was considered as the research object, and a modal analysis of the rotor system was performed using a mathematical model of mass concentration and via simulation using the ANSYS software. According to its natural frequency and vibration mode, the main conclusions obtained were as follows:

1. The four-stage centrifugal pump with a balance plate was calculated using lumped mass theory and simulated using ANSYS. The simulated natural frequencies were slightly higher than those obtained using the analytical method.

2. Based on an analysis of the differences in the calculation process and the results of the analytical and simulation methods, a modified mathematical model based on the concentrated mass method was proposed. The mass and flexural stiffness of the rotor system on the system modes were the main considerations in establishing the modified model. Additionally, a flexural stiffness correction coefficient was introduced.

3. The improved mathematical model reduced the errors in calculating the natural frequencies from 12.96%, 12.13%, 9.96%, 5.85%, and 8.74% to 2.45%, 1.56%, 0.65%, 5.34%, and 2.28%, respectively. A comparison of the mode shapes corresponding to each order of natural frequency revealed that the impeller located in the middle exhibited the greatest deformation. The optimization method for natural frequencies demonstrated better effectiveness in optimizing low-order mode shapes, whereas its effect on higher-order modes was less pronounced.

4. The optimization method of this mathematical model was applied to four-stage centrifugal pumps using back blades for axial-force balance. The errors in the calculated natural frequencies obtained using the analytical method decreased within 5% from 13.11%, 12.85%, 9.91%, and 7.2% to 3.7%, 3.86%, 0.57%, and 2.87%, respectively. This satisfies the operational requirements, thus further confirming the feasibility of the optimized model.

## Supporting information

**S1 Table. Calculation of flexibility influence coefficient of centrifugal pump rotor system.**
(XLSX)

**S1 File. Calculation of natural frequency and vibration mode.**
(PDF)

## Acknowledgments

We would like to thank Editage (www.editage.cn) for English language editing.

## Author Contributions

**Conceptualization:** Jingkuan Li, Hongbin Gao.

**Data curation:** Yanxia Wei, Hongbin Gao, Xiaomei Song.

**Formal analysis:** Jingkuan Li, Yanxia Wei, Hongbin Gao, Xiaomei Song.

**Funding acquisition:** Yanxia Wei, Xiaomei Song.

**Investigation:** Hongbin Gao.

**Methodology:** Jingkuan Li.

**Project administration:** Yanxia Wei, Hongbin Gao, Xiaomei Song.

**Resources:** Hongbin Gao.

**Software:** Jingkuan Li, Yanxia Wei.

**Visualization:** Zhuofan Jia.

**Writing – original draft:** Jingkuan Li, Xiaomei Song.

**Writing – review & editing:** Jingkuan Li, Hongbin Gao.

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
