## [Decision Letter · Decision Letter 0]

24 Apr 2024

PONE-D-24-10956An improved calculation method for dry modal analysis of four-stage centrifugal pump rotor system based on concentrated mass methodPLOS ONE

Dear Dr. Gao,

Thank you for submitting your manuscript to PLOS ONE. After careful consideration, we feel that it has merit but does not fully meet PLOS ONE’s publication criteria as it currently stands. Therefore, we invite you to submit a revised version of the manuscript that addresses the points raised during the review process.

 In addition to reviewers' comments, the authors are suggested to consider following points:(i) Abstract should be improved to avoid repetition of text(ii) Contours in Figures 3 and 6 are not of good quality. The values on color map are also not clear.(iii) The comparison with experimental studies should be included in the revised paper.(iv) The differences with the previous similar papers (Refs. 11 and 29) of the author need to discussed in detail.

We look forward to receiving your revised manuscript.

Kind regards,

Muhammad Shakaib, PhD

Academic Editor

PLOS ONE

When you resubmit, please ensure that you provide the correct grant numbers for the awards you received for your study in the ‘Funding Information’ section."

“The authors thank the support of Science and Technology Innovation project of Universities in Shanxi Province(No. 2023L016). We would like to thank Editage (www.editage.cn) for English language editing.”

Reviewers' comments:

Reviewer's Responses to Questions

**Comments to the Author**

1. Is the manuscript technically sound, and do the data support the conclusions?

Reviewer #1: Yes

Reviewer #2: Partly

Reviewer #3: Yes

2. Has the statistical analysis been performed appropriately and rigorously? 

Reviewer #1: Yes

Reviewer #2: Yes

Reviewer #3: No

3. Have the authors made all data underlying the findings in their manuscript fully available?

Reviewer #1: No

Reviewer #2: No

Reviewer #3: Yes

4. Is the manuscript presented in an intelligible fashion and written in standard English?

Reviewer #1: Yes

Reviewer #2: Yes

Reviewer #3: No

5. Review Comments to the Author

Reviewer #1: Reviewer Comments

In this paper, a simplified concentrated mass mathematical model was employed to enhance the accuracy of modal analysis using the concentrated mass method for a four-stage centrifugal pump rotor system. A modal analysis of a four-stage centrifugal pump rotor system with a balancing disc was conducted using both a simplified concentrated mass model and Ansys. By analyzing the differences between the two calculation results, correction coefficients based on the mass and flexural stiffness, which affect the modes of the system, were introduced. Subsequently, an improved mathematical model based on the concentrated mass method was proposed. However, the followings should be carefully addressed in the revision to be published in your journal.

1- The authors should be followed the instruction of the journal in all parts and sections in this manuscript.

2- Complete mathematic calculation model with all nomenclature missing. Please check the number of each section, equation, and chart.

3- The abstract needs more quantitative results. The abstract section is an important and powerful representation of the research. It is better that the results should be presented with the support of specified data. Please provide your contribution and work novelty.

4- The authors should indicate this technique to enhance system performance. Also, the author should add more references that discuss the effect of using this technique. It is recommended that the authors carry out wide analysis and comparison with the state-of-the-art studies.

5- Most tables and figures are needed improve the quality of all tables and figures.

6- Add references for all equations.

7- I would also expect to validate with two more experimental works available in the literature.

8- The literature review must be improved. Please highlight in the literature review the differences between previous papers and your paper. Please clearly indicate the knowledge gap and prove that it is a really not analyzed area of the field. Please indicate new approach / new methods in a comparison to the existing investigations (literature review should be extended and add below references). Evaluation and Investigation of Hydraulic Performance Characteristics in an Axial Pump Based on CFD and Acoustic Analysis. Investigation of the Main Flow Characteristics Mechanism and Flow Dynamics Within an Axial Flow Pump Based on Different Transient Load Conditions. Effect of Different Guide Vane Configurations on Flow Field Investigation and Performances of an Axial Pump Based on CFD Analysis and Vibration Investigation. Investigation of the Influence of Varying Operation Configurations on Flow Behaviors Characteristics and Hydraulic Axial-Flow Pump Performance. Investigation on the Characteristics of Internal Flow within Three-Dimensional Axial Pump Based on Different Flow Conditions. Experimental Diagnostic of Cavitation Flow in the Centrifugal Pump Under Various Impeller Speeds Based on Acoustic Analysis Method. Experimental and numerical investigations on the cavitation phenomenon in a centrifugal pump. Investigation of effect of pump rotational speed on performance and detection of cavitation within a centrifugal pump using vibration analysis.

9- You need to add error analysis of your results and add the error bars in your graphs to indicate your accuracy measurements.

10- Improve work justification. Also, add more analysis about velocity and pressure contours.

11- More quantitative conclusions should be presented. Please prepare additional comparisons, some percentage differences. There is a lack of quantitative conclusions which should contain main findings from the paper and highlight the new and high novelty and contribution of your work to the field.

12- Present the mathematical equation of the boundary conditions and initial condition.

13- I would also suggest including in the conclusion section but also in several other places in the manuscript discussion and comparison with findings from other authors with similar published research work.

14- The conclusion section on lacks in summative conclusions. The main results, novelty and academic contributions should be emphasized in this section. Moreover, are the results obtained in this paper really applicable in other similar researches?

15- In the discussion development, it is very important to emphasize points of agreement or disagreement between results in this work and others cited in references part of manuscript.

16- Authors should discuss limitations of the current study and possible improvements for future directions/research works. Authors are requested to check the reference format and correct some inconsistent formats.

17- Finally, I strongly recommend the author to read through the whole text and correct it to make it more reader-friendly.

Reviewer #2: In this article, a four-stage centrifugal pump rotor system model is established based on the basic of material mechanics and gives the inherent characteristics and modes of the structure through characteristic analysis. Meanwhile, the results are verified by comparing with Ansys and pointed out that the accuracy is obviously improved by the calculation method in the article.

There are several points that need to be improved and considered.

1. Some variables need to be indicated in italics to avoid misunderstandings.

2. The pictures in the article are blurry, a clearer picture will help to understand the contribution.

3. The author mention that this is an improved method, but just consider the division of the mass more carefully in the mass matrix and consider the stiffness of the impeller in the stiffness matrix after optimization. Calling it an improved method requires careful consideration.

4. In this article, only Ansys result is used for verification, if a comparison with experimental result is made will become more convincing.

5. The derivation and analysis generally is lack of innovation and the form of computation result is too simple which should be significantly improved.

6.

Reviewer #3: The authors studied a four disks rotor system. The natural frequency for the original and optimization system are studied and compared with the FEM simulation. For my opinion, the paper a little bit simple, and more like a report. At this situation, I think the paper can not be accepted.

Some comments are list as following.

1 In table 2, it’s hard to find the new nature frequency. It’s a simple copy of table 1.

2 What the key point for the authors to show the model shape again. I think the original, FEM and optimization system should have similar mode shape.

3 Usually, the first nature frequency should very close to the FEM and experiment results.

4 For a high-quality paper, I think a simple experimental verify is needed.

5 Novelty needs to be explicit in abstract.

6 Expand figure captions so figures are almost self-explanatory.

7 Broaden and update the literature review to better connect to the current effort in the field in the context of mechanical sciences; for papers like this one we expect no less than 40 journal papers including 15 recent ones to be critically discussed. Do not cite text books or manuals.

8 Conclusions should be stronger.

6. PLOS authors have the option to publish the peer review history of their article (what does this mean?). If published, this will include your full peer review and any attached files.

Reviewer #1: **Yes: **Ahmed Ramadhan Al-Obaidi

Reviewer #2: No

Reviewer #3: No

---

## [Author Response · Author response to Decision Letter 0]

28 May 2024

Responses to Reviewers’ Comments

Thank you for your valuable comments and suggestions regarding our manuscript. We have carefully considered each of the comment and performed the necessary revisions to improve the quality and clarity of our manuscript. These changes are indicated in red in the revised manuscript. Our responses to the reviewers’ comments are provided below:

Academic editor:

1.Abstract should be improved to avoid repetition of text.

Response: Thank you for your suggestion. We have revised the Abstract to eliminate this repetition. The revised version succinctly summarizes the research purpose, content, methodology, and conclusions and highlights the contributions and innovations of our study.

2.Contours in Figures 3 and 6 are not of good quality. The values on color map are also not clear.

Response: Thank you for highlighting this. We have considered your feedback in revising the manuscript. Specifically, we have improved the image quality by saving them in .tif format directly using the ANSYS software. This resulted in higher-quality images and clearer contours. Additionally, we have ensured that the values on the color map are clearly visible. These changes have been implemented in the revised map file.

3.The comparison with experimental studies should be included in the revised paper.

Response: Thank you for your suggestion. We appreciate your recommendation to include comparisons with experimental studies in the revised manuscript. However, we would like to emphasize that our focus is to propose an optimized mathematical model. In this study, we compared the results before and after optimization with ANSYS simulation results. We determined the effectiveness of the optimization model based on the reduction in the error values. Furthermore, we validated the applicability of the optimized model to a centrifugal pump rotor system with back blades. We believe that considering the factors affecting the errors in our study is reasonable. Our study can be used to improve the accuracy of analytical methods. Moreover, the ideas that you proposed provide directions for our future investigations.

4.The differences with the previous similar papers (Refs. 11 and 29) of the author need to discussed in detail.

Response: Thank you for your comment. Owing to the increased number of references, references 11 and 29 were updated to 31 and 32, respectively. The relevant content is briefly summarized in the Literature Review section of the manuscript. Reference [31] focused on a double-support four-stage centrifugal pump rotor system, where a simplified “four-segment four-concentration” mathematical model and an ANSYS model were used to solve the mechanical model of a D-type balanced-mass four-stage centrifugal pump rotor system. An improved mathematical model combining “five-segment four-concentration” and modal resistance-bending stiffness correction factors was proposed, and the effect of impeller-mass eccentricity on the modes was analyzed. The authors of [32] investigated a cantilever four-stage centrifugal pump rotor system using a simplified lumped-mass mathematical model and ANSYS to solve the natural frequencies of the mass-balanced rotor system. An optimized mathematical model integrating shaft system mass and modal stiffness correction factors was proposed, and the effect of impeller-mass eccentricity on the modes was analyzed. These two studies, as well as our current study, employed different structures of four-stage centrifugal pump rotor systems; hence, different mathematical models were established. Additionally, the factors considered during the optimization of the models varied owing to differences in the fixed methods and structures of the pump. Factors such as the unit-size division of the geometric model, material type, stiffness, and mass distribution contributed to significant errors in the natural frequencies. The focus of this study is the selection of lumped masses and the section moment of inertia. Based on the considerations above, an optimized mathematical model was established. The Literature Review section briefly describes this, with the modifications indicated in red.

Reviewer #1:

1-The authors should be followed the instruction of the journal in all parts and sections in this manuscript.

Response: Thank you for the comment. We acknowledge the importance of adhering to the instructions provided by the journal throughout the manuscript. We have carefully reviewed your comments and ensured that all the contents of the manuscript comply with the journal’s guidelines. The revised contents are indicated in red.

2-Complete mathematic calculation model with all nomenclature missing. Please check the number of each section, equation, and chart.

Response: Thank you for your comment. We have thoroughly reviewed the manuscript. We have revised the images to ensure clarity, renamed some of the image titles, and added references to the equations. All the changes are highlighted in red in the manuscript.

3-The abstract needs more quantitative results. The abstract section is an important and powerful representation of the research. It is better that the results should be presented with the support of specified data. Please provide your contribution and work novelty.

Response: Thank you for the suggestion. We have modified the abstract accordingly to highlight the key contributions of our study. In particular, we have summarized the research content, where we emphasized the use of an analytical method based on concentrated mass and the comparison of results with ANSYS simulation results. We presented specific numerical values to demonstrate the significant reduction in errors after optimization. The errors were within 5% in general, which satisfies the operational requirements and validates the feasibility of the optimization method. The focus of the study was the selection of concentrated mass and the moment of inertia of cross-sections, from which we derived an optimized mathematical model. Furthermore, we validated the applicability of the optimization method to another rotor system, which enabled the assessment of the system's vibration characteristics under specific operating conditions, thereby providing important insights for pump design and optimization. The rotor system with a balance disc fixed by a simply supported beam used in this study has not been reported previously. The modified sections of the Abstract are highlighted in red.

4-The authors should indicate this technique to enhance system performance. Also, the author should add more references that discuss the effect of using this technique. It is recommended that the authors carry out wide analysis and comparison with the state-of-the-art studies.

Response: Thank you for your comment. We have revised the Literature Review section to include more references. Additionally, we have discussed the application of the lumped-mass method in modal analysis, the effects of different structural materials on modal behavior, and the use of modal analysis in structural damage detection. The added information highlight the importance of modal analysis and its significance in optimizing the structural design of centrifugal pumps, improving pump performance and efficiency, reducing noise and vibration, and ensuring safe equipment operation. Changes to the manuscript are highlighted in red.

5-Most tables and figures are needed improve the quality of all tables and figures.

Response: Thank you for your comment regarding the quality of the tables and graphics in the manuscript. We have scrutinized every table and graph and improved their quality accordingly. Specifically, we improved the resolution of the image, adjusted its file format to enhance clarity, ensured that all labels and annotations were accurately matched, and renamed some images. These improvements have been incorporated into the revised manuscript, along with updated figure numbers.

6-Add references for all equations.

Response: Thank you for your comment. We have added references to all the equations in the manuscript. The equations were derived primarily from References [34] and [35], and the added citations are indicated in red.

7-I would also expect to validate with two more experimental works available in the literature.

Response: Thank you for your comment. We agree that additional studies will help us better understand the details of these interactions and enhancements. However, we would like to emphasize that our primary focus is to propose an optimized mathematical model to improve the accuracy of the concentrated-mass method. By comparing the values obtained from the analytical calculations with those from the ANSYS simulation, the significant reduction in errors validates the effectiveness of the optimized model. Thus, the proposed approach can facilitate future investigations.

8-The literature review must be improved. Please highlight in the literature review the differences between previous papers and your paper. Please clearly indicate the knowledge gap and prove that it is a really not analyzed area of the field. Please indicate new approach / new methods in a comparison to the existing investigations (literature review should be extended and add below references).

Response: Thank you for your comment. We have revised the Literature Review section as follows:

(1) We provided an overview of studies pertaining to the modal characteristics of rotor systems conducted by scholars both domestically and abroad.

(2) We discussed the application of modal-analysis methods in structural damage detection.

(3) We elaborated on the effects of different structural materials on the modes.

(4) We highlighted studies by foreign scholars pertaining to the application of the centralized-mass method used in this study to other fields.

By performing a literature review, we identified the cutting-edge research directions and application areas of modal analysis. Currently, studies regarding the design optimization of mathematical models for centrifugal-pump rotor systems are limited. This paper delves into this aspect to improve the accuracy of modal calculations and provide a foundation for the structural design of centrifugal pumps and their vibration and noise reduction. Additionally, we have included the references you mentioned in the revised manuscript (see References [3], [4], [5], [10], [11], [16], [17], and [42]). Modifications to the literature review and references are highlighted in red.

9-You need to add error analysis of your results and add the error bars in your graphs to indicate your accuracy measurements.

Response: Thank you for your comment. We have added two error-comparison diagrams to the revised manuscript, as shown in <Response to Reviewers>.

Comparison of analytical calculations and optimization results

Based on the error comparison presented, we clearly observed a significant decrease in the optimized error values. The error-comparison charts clearly demonstrated the effectiveness of the optimized mathematical model in improving the calculation accuracy of the analytical method.

Comparison between analytical calculation and optimization results of back-vane centrifugal pump

Based on a comparison between the analytical calculation and optimization results, the errors in the analytical calculation values decreased from 13.11%, 12.85%, 9.91%, and 7.2% to 3.7%, 3.86%, 0.57%, and 2.87%, respectively, thus indicating the effectiveness of the optimized mathematical model.

10-Improve work justification. Also, add more analysis about velocity and pressure contours.

Response: Thank you for your comment. We understand the importance of velocity and pressure contours in certain scenarios. However, our primary focus is to propose an optimized mathematical model to improve the accuracy of the concentrated-mass method. This involves the application of both the concentrated-mass analytical method and ANSYS simulation, through which the natural frequencies and mode shapes can be determined. Based on the numerical results, we analyzed the desired modes and deformations. Additionally, we prioritized modal analysis as it aligns with our research objectives and investigation scope. We discovered that the velocity cloud diagram can show the velocity distribution of a fluid flowing through a pipeline or device, whereas the pressure cloud diagram can show the pressure level of a fluid at different positions. However, our study is based on the dry mode; hence, the above-mentioned diagrams are not relevant. Nevertheless, we will ensure that our study is fully demonstrated and will consider providing additional analyses of velocity and pressure contours in our future fluid–solid coupled studies.

11-More quantitative conclusions should be presented. Please prepare additional comparisons, some percentage differences. There is a lack of quantitative conclusions which should contain main findings from the paper and highlight the new and high novelty and contribution of your work to the field.

Response: Thank you for your comment. We have revised the Conclusion section accordingly. Specifically, we have succinctly summarized the main contents of our paper and outlined the key conclusions of our study. By performing numerical comparisons, we demonstrated a significant reduction in errors, i.e., errors within 5% in general after optimization. The errors are primarily due to the improper selection of concentrated-mass segments and the incomplete consideration of the cross-sectional moment of inertia factors. Based on these findings, we derived correction formulas for the optimized mathematical model. Furthermore, we validated the feasibility of the optimization model. The proposed optimization model is the focus and novelty of our study. Enhancing the accuracy of the concentrated-mass method facilitates the structural design of centrifugal pumps by avoiding noise and vibration. This is important for the stable operation of centrifugal pumps and cost reduction. Modifications to the conclusions are indicated in red.

12-Present the mathematical equation of the boundary conditions and initial condition.

Response: Thank you for your comment. The mathematical model used in this study applies the equation of motion of a multi-degree-of-freedom linear structural system in <Response to Reviewers>.

Modal analysis considers the inherent characteristics of the system; therefore, undamped free vibration is analyzed, i.e. in <Response to Reviewers>. The free-vibration equation of the system can be expressed as: in <Response to Reviewers> . To solve the mode in ANSYS, one must establish the model, partition the grid, apply constraint conditions, and finally solve the output results of the model. Grid partitioning significantly affects the results of modal analysis. The finer the grid, the more accurate are the calculation results. In this study, the entire model was selected for geometric size adjustment. The unit size was 7 mm, and the entire model was segmented into 183431 nodes and 102154 units. A constraint condition was applied to simulate a simply supported beam; one end was the fixed hinge support, and the other end was the sliding hinge support. The left end of the simply supported beam was selected to constrain the displacements in the X- and Z-directions, whereas the right end was selected to constrain the displacement in the Z-direction. The entire beam was selected to constrain the displacement in the X-direction.

13- I would also suggest including in the conclusion section but also in several other places in the manuscript discussion and comparison with findings from other authors with similar published research work.

Response: Thank you for your comment. The manuscript has been revised to include a brief comparison of two articles by the same authors in the Literature Review section. Reference [31] focuses on a double-supported four-stage centrifugal pump rotor system, where a simplified “four-segment four-concentration” model and an ANSYS model were used to solve the mechanical model of a D-type balanced-mass four-stage centrifugal pump rotor system. Additionally, an improved mathematical model combining “five-segment four-concentration” and a modal resistance-bending stiffness-correction coefficie

---

## [Decision Letter · Decision Letter 1]

11 Jun 2024

Improved calculation method for dry modal analysis of four-stage centrifugal-pump rotor system based on concentrated-mass method

PONE-D-24-10956R1

Dear Dr. Gao,

We’re pleased to inform you that your manuscript has been judged scientifically suitable for publication and will be formally accepted for publication once it meets all outstanding technical requirements.

Kind regards,

Muhammad Shakaib, PhD

Academic Editor

PLOS ONE

Additional Editor Comments (optional):

Reviewers' comments:

Reviewer's Responses to Questions

**Comments to the Author**

1. If the authors have adequately addressed your comments raised in a previous round of review and you feel that this manuscript is now acceptable for publication, you may indicate that here to bypass the “Comments to the Author” section, enter your conflict of interest statement in the “Confidential to Editor” section, and submit your "Accept" recommendation.

Reviewer #1: (No Response)

Reviewer #2: All comments have been addressed

Reviewer #3: (No Response)

2. Is the manuscript technically sound, and do the data support the conclusions?

Reviewer #1: Yes

Reviewer #2: Yes

Reviewer #3: Yes

3. Has the statistical analysis been performed appropriately and rigorously? 

Reviewer #1: Yes

Reviewer #2: Yes

Reviewer #3: Yes

4. Have the authors made all data underlying the findings in their manuscript fully available?

Reviewer #1: Yes

Reviewer #2: Yes

Reviewer #3: Yes

5. Is the manuscript presented in an intelligible fashion and written in standard English?

Reviewer #1: Yes

Reviewer #2: Yes

Reviewer #3: Yes

6. Review Comments to the Author

Reviewer #1: The followings should be carefully addressed in the revision to be published in your journal.

1- The abstract needs more quantitative results. The abstract section is an important and powerful representation of the research. It is better that the results should be presented with the support of specified data. Please provide your contribution and work novelty.

2- The authors should indicate this technique to enhance system performance.

3- Add references for all equations.

4- Improve work justification. Also, add more analysis about velocity and pressure contours.

5- More quantitative conclusions should be presented.

Reviewer #2: Review of "Improved calculation method for dry modal analysis of four-stage centrifugal-pump rotor system based on concentrated-mass method". This paper improve the accuracy of modal analysis for a four-stage centrifugal-pump rotor system with a balancing disc based on the concentrated-mass analytical method, and a simplified concentrated mass mathematical model and an ANSYS simulation model are established. The results from these two models are reasonable now. On the whole, the authors have address the former comments.

Reviewer #3: The authors had done a good work now. By considering the pre-review reports, I think this paper can be accepted at this situation.

7. PLOS authors have the option to publish the peer review history of their article (what does this mean?). If published, this will include your full peer review and any attached files.

Reviewer #1: **Yes: **Ahmed Ramadhan Al-Obaidi

Reviewer #2: No

Reviewer #3: No

---

## [Editor Report · Acceptance letter]

20 Jun 2024

PONE-D-24-10956R1 

PLOS ONE

Dear Dr. Gao, 

I'm pleased to inform you that your manuscript has been deemed suitable for publication in PLOS ONE. Congratulations! Your manuscript is now being handed over to our production team.

Kind regards, 

on behalf of

Dr. Muhammad Shakaib 

Academic Editor

PLOS ONE